# Glycated albumin as a diagnostic tool in diabetes: An alternative or an additional test?

Fernando Chimela Chume[1,2], Mayana Hernandez Kieling[1], Priscila Aparecida Correa Freitas[1,3], Gabriela Cavagnolli[4], Joíza Lins Camargo [1,5]*

1 Graduate Program in Medical Sciences: Endocrinology, Universidade Federal do Rio Grande do Sul Porto Alegre, Porto Alegre–RS, Brazil, 2 Universidade Zambeze, Beira, Mozambique, 3 Laboratory Diagnosis Division, Clinical Biochemistry Unit, Hospital de Clinicas de Porto Alegre (HCPA), Porto Alegre–RS, Brazil, 4 Centro Universitário FSG, Caxias do Sul—RS, Brazil, 5 Endocrinology Division and Experimental Research Centre, Hospital de Clinicas de Porto Alegre, Porto Alegre–RS, Brazil

* jcamargo@hcpa.edu.br

**Data Availability Statement:** Data cannot be shared publicly due to ethical restrictions. The use of the data for further research needs to be approved by the Research Ethics Committee of the Hospital de Clinicas de Porto Alegre. Requests to

## Abstract

### Introduction

Studies have revealed that glycated albumin (GA) is a useful alternative to $HbA_{1c}$ under conditions wherein the latter does not reflect glycaemic status accurately. Until now, there are few studies with non-Asians subjects that report on the validity of GA test in diagnosis of type 2 diabetes mellitus (DM). Thus, the aim of this study was to assess the clinical utility of GA in diagnosis of DM.

### Materials and methods

This diagnostic test accuracy study was performed in 242 Brazilian individuals referred for OGTT in a tertiary university hospital. ROC curves were used to access the performance of GA and $HbA_{1c}$ in the diagnosis of DM by oral glucose tolerance test (OGTT).

### Results

OGTT, $HbA_{1c}$ and GA were performed in all 242 participants (40.5% male, age 54.4 ± 13.0 years [mean ± SD], body mass index 28.9 ± 6.3 kg/m$^2$). DM by OGTT was detected in 31.8% of individuals. The equilibrium threshold value of GA $\geq$14.8% showed sensitivity of 64.9% and specificity of 65.5% for the diagnosis of DM. The AUC for GA [0.703 (95% CI 0.631–0.775)] was lower than for $HbA_{1c}$ [0.802 (95% CI 0.740–0.864)], p = 0.028. A GA value of 16.8% had similar accuracy for detecting DM as defined by $HbA_{1c}$ $\geq$6.5% (48 mmol/mol) with sensitivity of 31.2% and specificity of 93.3% for both tests. However, GA detects different subjects from those detected by $HbA_{1c}$ and OGTT.

### Conclusions

GA detected different individuals with DM from those detected by $HbA_{1c}$, though it showed overall diagnostic accuracy similar to $HbA_{1c}$ in the diagnosis of DM. Therefore, GA should be used as an additional test rather than an alternative to $HbA_{1c}$ or OGTT and its use as the sole DM diagnostic test should be interpreted with caution.

access the data may be submitted to corresponding author (contact via jcamargo@hcpa.edu.br) or contact Human Research Ethics Committee of the Hospital de Clinicas de Porto Alegre (HCPA) via cep@hcpa.edu.br.

**Funding:** This work was supported by the Research Incentive Fund (FIPE) of the Hospital de Clínicas de Porto Alegre (HCPA). FCC received scholarship from Ministry of Science and Technology, Higher Education and Professional Technician (MCTESTP) of the Republic of Mozambique. MKH received a undergraduate scholarship from Conselho Nacional de Desenvolvimento Científico e Tecnológico (CNPq). The funders had no role in study design, data collection and analysis, decision to publish, or preparation of the manuscript.

**Competing interests:** The authors have declared that no competing interests exist.

**Abbreviations:** DM, type 2 diabetes mellitus; FPG, fasting plasma glucose; OGTT, oral glucose tolerance test; HbA$_{1c}$, glycated haemoglobin; GA, glycated albumin; GSP, glycated serum proteins; HCPA, Hospital de Clinicas de Porto Alegre; LR, likelihood ratios; ROC, receiver operating characteristic; AUC, area under the ROC curve; WC, waist circumference; 2hPG, 2-h plasma glucose after a 75-g OGTT.

## Introduction

Despite being largely preventable, the worldwide increase in type 2 diabetes mellitus (DM) is becoming a major health concern. It has been estimated that globally as many as 212.4 million people or half (50.0%) of all people aged 20–79 years old with DM are unaware of their disease [1]. Any improvement in the identification of hyperglycaemia will be of significant impact because delays in diagnosis and treatment may increase the incidence of cardiovascular outcomes and all-cause mortality related to this disease [2]. At present DM may be diagnosed based on plasma glucose criteria, either by fasting plasma glucose (FPG) or 2-h plasma glucose (2hPG) after a 75-g oral GTT (OGTT) or HbA$_{1C}$ criterion, all tests are equally appropriate [3].

Although OGTT measurement is still a standard recommendation for DM diagnosis, this method is onerous, time-consuming and requires two blood samples. In contrast, the sole use of FPG measurement in DM screening will fail to diagnose those subjects presenting only with 2hPG $\geq$11.1 mmol/L ($\geq$200 mg/dL). Moreover, HbA$_{1c}$, which is considered the reference standard for monitoring long-term glycaemic control in subjects with DM, is also a primary diagnostic tool for DM. HbA$_{1c}$ has several advantages compared with the FPG and OGTT, including greater convenience (fasting is not required), higher pre-analytical stability, and less day-to-day variations during stress and illness [3]. However, HbA$_{1c}$ is not suitable for conditions with altered blood red cell turnover, such as some haemoglobinopathies, thalassemia, chronic kidney disease and haemolytic anaemia [4]. Furthermore, the presence of haemoglobin variants (e.g. HbS trait, HbC trait), elevated foetal haemoglobin (HbF) and chemically modified derivatives of haemoglobin (e.g. carbamylated Hb in patients with renal failure) can interfere either positively or negatively with the HbA$_{1c}$ measurement and consequently adversely affect the interpretation of HbA$_{1c}$ results [4–6]. Therefore, it is important to consider alternative procedures in the diagnosis of DM.

Glycated albumin (GA) is a ketamine produced by binding of albumin and glucose by a nonenzymatic glycation reaction [7]. It reflects short-term mean glycaemic values (2–3 weeks) due to the shorter half-life of serum albumin, rather than 2–3 months mean glycaemic values observed in HbA$_{1c}$ [8]. Similar to HbA$_{1c}$, GA correlates with diabetic complications such as retinopathy, chronic kidney disease, peripheral neuropathy, cardiovascular disease, and even death [9–11]. Additionally, GA is haemoglobin/erythrocyte independent, consequently, measurement of GA is not influenced by anaemia or other conditions considered potential factors that can affect the interpretation of HbA$_{1c}$ results [7, 8]. Besides, evidences suggest that GA is a better glycaemic indicator than HbA$_{1c}$ in diabetic subjects on haemodialysis [7].

Although data about GA performance in diagnosis and screening of DM have been available in Asian populations [12–15], limited data exists in other populations [16–18]. We hypothesized that GA may be used in the diagnosis of DM and in clinical conditions where the HbA$_{1c}$ test does not accurately reflect blood glucose concentrations GA may be an alternative marker. Therefore, the current study was designed to assess the clinical utility of GA in screening and diagnosis of DM in Brazilian individuals.

## Material and methods

### Study design

We conducted a cross-sectional study of diagnostic accuracy and reported corresponding results according to Standard for Reporting Diagnostic Accuracy (STARD) statement [19]. The study flow diagram is shown in Fig 1.

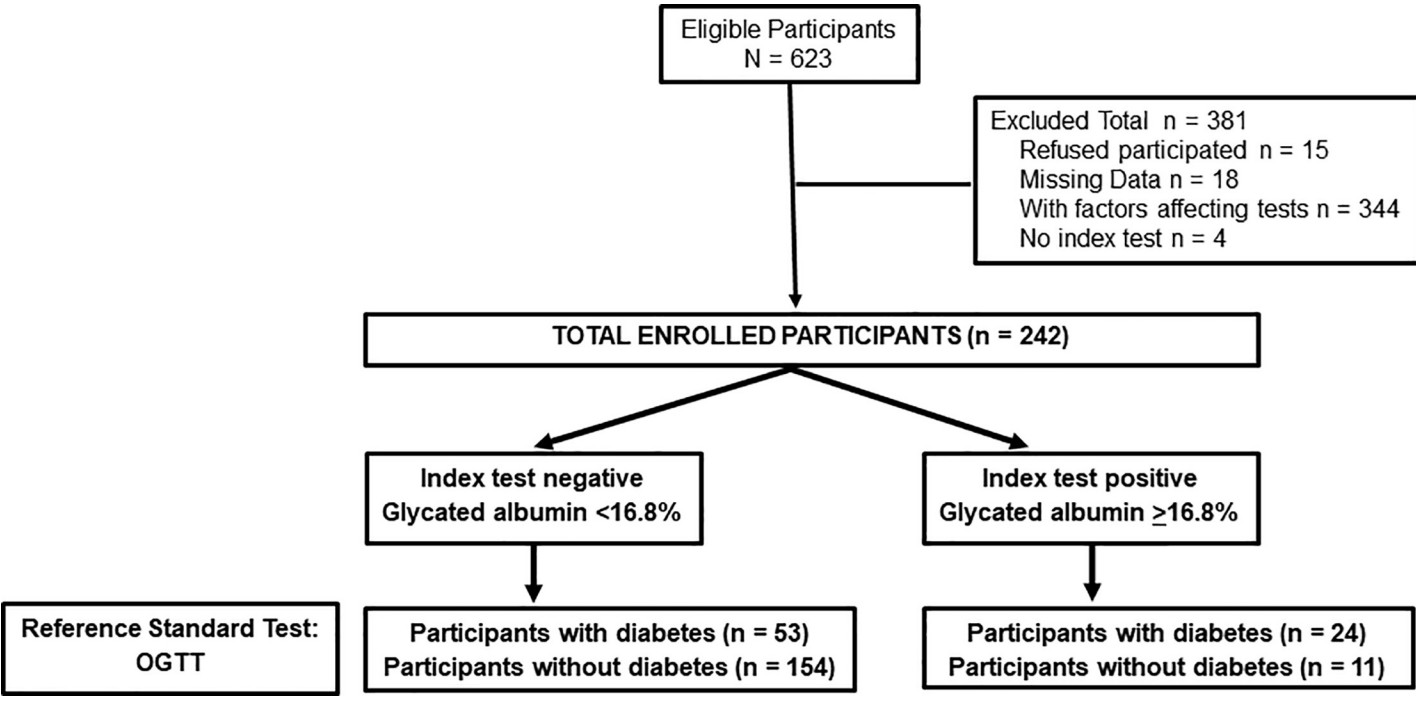

**Fig 1. Study flow diagram.** OGTT, oral glucose tolerance test.

## Participant selection

Outpatients older than 18 years referred to the Hospital de Clinicas de Porto Alegre (HCPA) between August 2008 and August 2017 to perform OGTT were consecutively invited to participate in this study. Subjects who accepted the invitation completed a questionnaire, underwent a physical examination and received blood tests. Serum sample was stored at -80˚C for GA measurement. The stability of the GA assay in long-term stored specimens has already been evaluated [20].

Study exclusion criteria were: albumin levels <3.0 g/dl; subjects with established diagnosis of DM or who were receiving anti-diabetic medication; pregnant women; presence of anaemia, hemoglobinopathy, recent transfusion, rheumatic disorder, hepatic cirrhosis, nephrotic syndrome, chronic kidney disease, untreated thyroid dysfunction, and/or Cushing syndrome, since these disorders are known to influence values of GA and/or $HbA_{1c}$.

Each participant provided a written informed consent. This study was reviewed and approved by the Research Ethics Committee of the Hospital de Clinicas de Porto Alegre (GPPG 080321 and 160448).

Glycaemic status was defined according to American Diabetes Association criteria [3]. DM was defined by: *(a)* FPG ≥7.0 mmol/L (≥126 mg/dL) and/or *(b)* 2hPG ≥11.1 mmol/L (≥200 mg/dL) during an OGTT and/or *(c)* $HbA_{1c}$ ≥6.5% (≥48 mmol/mol) for descriptive purposes. With the intention of a diagnostic accuracy study, OGTT [FPG ≥7.0 mmol/L (≥126 mg/dL) and/or 2hPG ≥11.1 mmol/L (≥200 mg/dL)] was used as reference standard test; $HbA_{1c}$ and GA were used as index tests.

## Laboratorial methods

All subjects underwent a standard 75g OGTT after an overnight fast of at least 8 hours. Blood samples for glucose determination were collected by venepuncture into tubes containing

sodium fluoride at fasting and at 2-hour after 75g glucose oral intake. Plasma glucose concentrations were measured by colorimetric enzymatic method in the biochemistry automated analyser Cobas® c702 (Roche Diagnostics, Germany).

HbA$_{1c}$ were measured in K2EDTA-anticoagulated whole blood by high performance liquid chromatography (HPLC) using VARIANT II™ System (BioRad Laboratories, Hercules, CA, USA). This HbA$_{1c}$ assay is certified by the National Glycohemoglobin Standardization Programme, aligned to the DCCT assay and it is also standardized by International Federation of Clinical Chemistry [21]. Analytical inter-assay coefficient of variation in our lab was <3.0% [22].

Fasting serum samples were stored at -80˚C until it was used for measurement of GA. GA were determined by an enzymatic method (GlycoGap®, Diazyme Laboratories, Poway, CA) in the automated analyser Cobas® c702 (Roche Diagnostics, Germany). This method was previously validated in our lab and the intra-assay repeatability was 3.5% [22]. Total albumin was measured with bromocresol green colorimetric method. GlycoGap® GA assay quantifies the total of glycated serum proteins (GSP, µmol/L), which are converted to percent of GA by the following conversion equation: GA (%) = {[GSP (µmol/L) x 0.182 + 1.97]/total albumin (g/dL)} + 2.9 [22]. Previous results showed that the Diazyme method correlates well with the Lucica GA-L assay, a specific GA assay used in Asian and Europe, with a small bias, supporting the equivalence between GSP and GA [23].

Serum creatinine was measured by Jaffé colorimetric method, triglycerides, total cholesterol by enzymatic assays, both using Cobas® c702 analyser (Roche Diagnostic, Mannheim, Germany). Haemoglobin and haematocrit were assayed by routine techniques.

Body mass index (BMI) was calculated by dividing body weight (kg) by the square of body height (m). The waist circumference was measured midway between the lowest rib and the iliac crest in a standing position. Systolic blood pressure and diastolic blood pressure were measured on the right arm, in the sitting position, with an automated sphygmomanometer (HEM-780, Omron Healthcare, Kyoto, Japan) after at least 5-minute rest. Smoking and drinking habit (current, past or never), and ethnicity was determined by self-report.

## Statistical analysis

Unless otherwise stated, data are presented as mean ± standard deviation (SD) for continuous variables and as percentages for categorical variables. Group comparisons were analysed by Student's t-test, Fisher's exact test and the Chi-square test as appropriate. For clinical and laboratory descriptive purposes, individuals with and without DM were divided using ADA OGTT criteria. Relationships among variables were explored using Spearman's correlation coefficients and regression models. Receiver operating characteristic (ROC) curves were used to access the performance of GA and HbA$_{1c}$ in the diagnosis of DM by OGTT as the reference test. Also ROC curve was created to evaluate the performance of GA using OGTT and/or HbA$_{1c}$ as DM diagnostic reference test. Areas under the curves (AUC) of GA and HbA$_{1c}$ were compared by DeLong's test. The optimal cut-off for serum GA was derived from the ROC curve with the shortest distance to sensitivity and specificity with maximum value of the Youden index. Combining sensitivity and specificity, we calculated likelihood ratios (LR) for different cut-off points. The LR+ was calculated by dividing the sensitivity of the test by 1 −specificity (Sensitivity/1 − specificity), while LR− of a test can be calculated by dividing 1 − sensitivity by specificity (1 − Sensitivity/Specificity) [24]. The first cut-off point of GA in the ROC curve with specificity over 90%DM was chosen as the criterion for diagnosis of DM. To demonstrate the clinical applicability of the test, we combined likelihood ratios with pre-test probability of the disease to estimate post-test probability using Fagan's nomogram [25]. Venn

diagram was used to present the number of individuals with DM diagnosed by each test and overlaps.

The IBM SPSS software for Windows, version 20.0 (Statistical Package for Social Sciences—Professional Statistics, IBM Corp, Armonk, USA) and MedCalc, version 19.1 (MedCalc software, Ostend, Belgium) were used for data analysis. P values 0.05 were considered significant.

## Results

A total of 242 participants were enrolled in the present study, of those 144 (69.5%) were women. One hundred ninety-five (80.2%) subjects self-reported European ancestry (mainly of Portuguese, German, Italian and Spanish descent). Participants presented mean age of 54.4 years (± 13.0) and values for GA, FPG, 2hPG, and $HbA_{1c}$ of 14.9 ± 2.2%, 6.2 ± 1.2 mmol/l (112 ± 21 mg/dL), 9.2 ± 4.1 mmol/l (165 ± 73 mg/dL), 5.79 ± 0.79% (40 ± 8.6 mmol/mol), respectively. GA values were not normally distributed [median 14.5% (GA $_{minimum}$ 8.2%, GA $_{maximum}$ 26.9%)]. Based on glucose criteria for the OGTT, DM was detected in 31.8% (77/242). $HbA_{1c}$ ≥6.5% (48 mmol/mol) identified 33 individuals with DM (13.6%), of those subjects 24 were also diagnosed with DM by OGTT. Based on both tests, a total of 86 participants had diagnosis of DM (35.5%).

The clinical and laboratory characteristics of all individuals are shown in Table 1. Individuals with DM diagnosed by ADA OGTT criteria, compared to the group without DM, were older and had higher values of GA, FPG, 2hPG and $HbA_{1c}$. There were no significant differences in BMI and HDL. On the other hand, subjects with DM had higher values of total cholesterol, triglyceride and LDL. Additionally, the ethnic difference between groups was not accessed due to small sample size.

The correlations between GA and factors potentially associated with the measurement of serum GA in all participants are presented in S1 Table. GA and age were positively correlated (r = 0.294, p <0.001). GA concentrations increased by 0.44% per decade (GA = 12.503 + 0.044 x age). GA was inversely correlated with triglyceride (r = - 0.197, p < 0.001). For every 10 mg/dL increase in serum triglyceride, GA decreased by 0.04% (GA = 15.623–0.004 x [triglyceride]). However, in those participants recently diagnosed with DM, these correlations were not significant [age (r = 0.101, p = 0.380) and triglyceride (r = -0.020, p = 0.429)]. Whereas $HbA_{1c}$ was positively correlated with BMI, waist circumference, total cholesterol, triglyceride and low-density lipoprotein cholesterol (LDL); GA was negatively correlated with BMI, WC, total cholesterol, triglyceride and LDL, though most of these correlations were not significant (S1 Table).

ROC curves comparing the performance of GA and $HbA_{1c}$ in the diagnosis of DM by OGTT as the reference test are presented in Fig 2. The AUC for GA in the diagnosis of DM by the OGTT was lower than for $HbA_{1c}$ (p = 0.028), with values of 0.703 (95% CI 0.631–0.775) and 0.802 (95% CI 0.740–0.864), for GA and $HbA_{1c}$, respectively. The equilibrium cut-off value for GA was 14.8%; sensitivity and specificity for GA in this cut point were 64.9% and 65.5%, respectively. GA ≥14.8% yielded LR+ and LR- of 1.88 and 0.54, respectively (Table 2). Inferring in our population a pre-test probability of 9.0% for DM [1] and considering GA ≥14.8% as DM diagnostic criterion, after a positive test (GA ≥14.8%) the post-test probability for DM would increase to 16%, while a negative test (GA <14.8%) would decrease the post-test probability for DM to 5%. In this study, using the equilibrium point of GA as the criterion for diagnosis of DM (GA <14.8%), 50 subjects with DM would have a true positive diagnosis; however, 27 subjects with DM and 57 subjects without DM would be falsely diagnosed.

We also evaluated the performance of GA using OGTT and/or $HbA_{1c}$ as DM diagnostic reference test. There was no relevant change in the AUC of GA compared to the one obtained

**Table 1. Clinical and laboratory characteristics of the study participants divided by subjects with and without DM using ADA OGTT criteria.**

| | Total | Without DM | DM | P |
|---|---|---|---|---|
| n | 242 | 165 | 77 | |
| Age (years) | 53.4 ± 13.4 | 56.8 ± 11.9 | 58.5 ± 11.5 | 0.056 |
| Sex (male/female) | 98/144 | 75/90 | 23/54 | 0.025 |
| Ancestry | | | | 0.147 |
| European, n (%) | 195 (80.5) | 135 (80.5) | 62 (80.5) | |
| African, n (%) | 28 (11.6) | 16 (9.8) | 12 (15.6) | |
| Other ancestry, n (%) | 19 (7.9) | 16 (9.8) | 3 (3.9) | |
| BMI (kg/m2) | 28.9 ± 6.3 | 28.6 ± 6.4 | 29.6 ±+ 6.3 | 0.271 |
| WC (cm) | 99.26 ± 13.38 | 98.8 ± 14.0 | 100.3 ± 11.9 | 0.429 |
| SBP (mm Hg) | 131.69 ± 16.74 | 130 ± 16 | 139 ± 17 | 0.016 |
| DBP (mm Hg) | 80.38 ± 12.72 | 79 ± 12 | 85 ± 14 | 0.051 |
| Family history of DM, n (%) | 122 (51.3) | 73 (45.1) | 49 (64.5) | 0.005 |
| Hypertension, n (%) | 146 (60.8) | 91 (55.8) | 55 (71.4) | 0.021 |
| Hypertension treatment, n (%) | 139 (57.4) | 87 (53.4) | 52 (67.5) | 0.038 |
| Total cholesterol (mmol/l) | 4.9 ± 1.1 | 4.8 ± 1.0 | 5.1 ± 1.2 | 0.017 |
| Triglycerides (mmol/l) | 1.9 ± 1.2 | 1.7 ± 1.0 | 2.2 ± 1.4 | 0.003 |
| HDL (mmol/l) | 1.2 ± 0.4 | 1.2 ± 0.9 | 1.2 ± 0.3 | 0.934 |
| LDL (mmol/l) | 3.7 ± 1.1 | 3.5 ± 1.1 | 3.9 ± 1.2 | 0.021 |
| Serum Creatinine (μmol/l) | 73.2 ± 19.4 | 73.4 ± 17.7 | 76.0 ± 17.7 | 0.334 |
| Serum albumin (g/l) | 44.0 ± 4.0 | 44.0 ± 4.0 | 44.0 ± 4.0 | 0.773 |
| Haemoglobin (g/l) | 13.8 ± 1.4 | 13.8 ± 1.4 | 13.8 ± 1.4 | 0.942 |
| FPG (mmol/l) | 6.2 ± 1.2 | 5.8 ± 0.6 | 7.2 ± 1.4 | < 0.001 |
| 2hPG (mmol/l) | 9.2 ± 4.1 | 7.2 ± 2.0 | 13.4 ± 4.1 | < 0.001 |
| $HbA_{1c}$ (%) (mmol/mol) | 5.8 ± 0.8 40.0 ± 8.6 | 5.5 ± 0.6 38.0 ± 6.6 | 6.3 ± 0.9 43.0 ± 12.0 | < 0.001 |
| GA (%) | 14.91 ± 2.2 | 14.4 ± 1.8 | 15.9 ± 2.6 | < 0.001 |

Mean ± SD and for continuous variables. ADA, American Diabetes Association; BMI, body mass index; WC, waist circumference (cm); SBP, systolic blood pressure; DBP diastolic blood pressure; HDL, serum high density lipoprotein cholesterol; LDL, serum low density lipoprotein cholesterol; FPG, fasting plasma glucose; 2hPG, plasma glucose 2 h after oral glucose; $HbA_{1c}$, glycated haemoglobin; GA, glycated albumin; OGTT, oral glucose tolerance test; DM, type 2 diabetes mellitus.

when OGTT solely is considered as diagnostic reference test [0.708 (95% CI 0.639–0.776)] versus [0.703 (95% CI 0.631–0.775)], respectively. The optimal cut-off value for serum GA, when OGTT and/or $HbA_{1c}$ are reference was 14.7% (sensitivity 64.0% and specificity 64.1%) versus 14.8% (sensitivity 64.9% and specificity 65.5%) when OGTT alone is reference.

GA value of 16.6% was the first point in the ROC curve presenting specificity higher than 90%. However, the cut-off of 16.8% had similar performance for detecting DM as defined by $HbA_{1c}$ ≥6.5% (≥48 mmol/mol) with sensitivity of 31.2% and specificity of 93.3% and presented LR+ of 4.68 and LR- of 0.74 (Table 2). Therefore, considering a pre-test probability of 9.0% [1], after a positive test (GA ≥16.8%), the post-test probability for DM would increase to 32%, while a negative test (GA <16.8%) would decrease the post-test probability for DM to 7% (Fig 3). In our study group, considering this point (GA ≥16.8%), the number of truly negative subjects would increase to 154, the number of false positive results would be reduced to 11, however also would reduce the truly positive results to 24. However, it should be noted that GA, $HbA_{1c}$ and OGTT do not necessarily detect DM in the same individuals (Fig 4). Among 77 subjects diagnosed with DM by OGTT, only 11 were identified as DM subjects by both GA ≥16.8% and $HbA_{1c}$ ≥6.5%. Thirteen of the remaining 66 subjects were identified only by GA

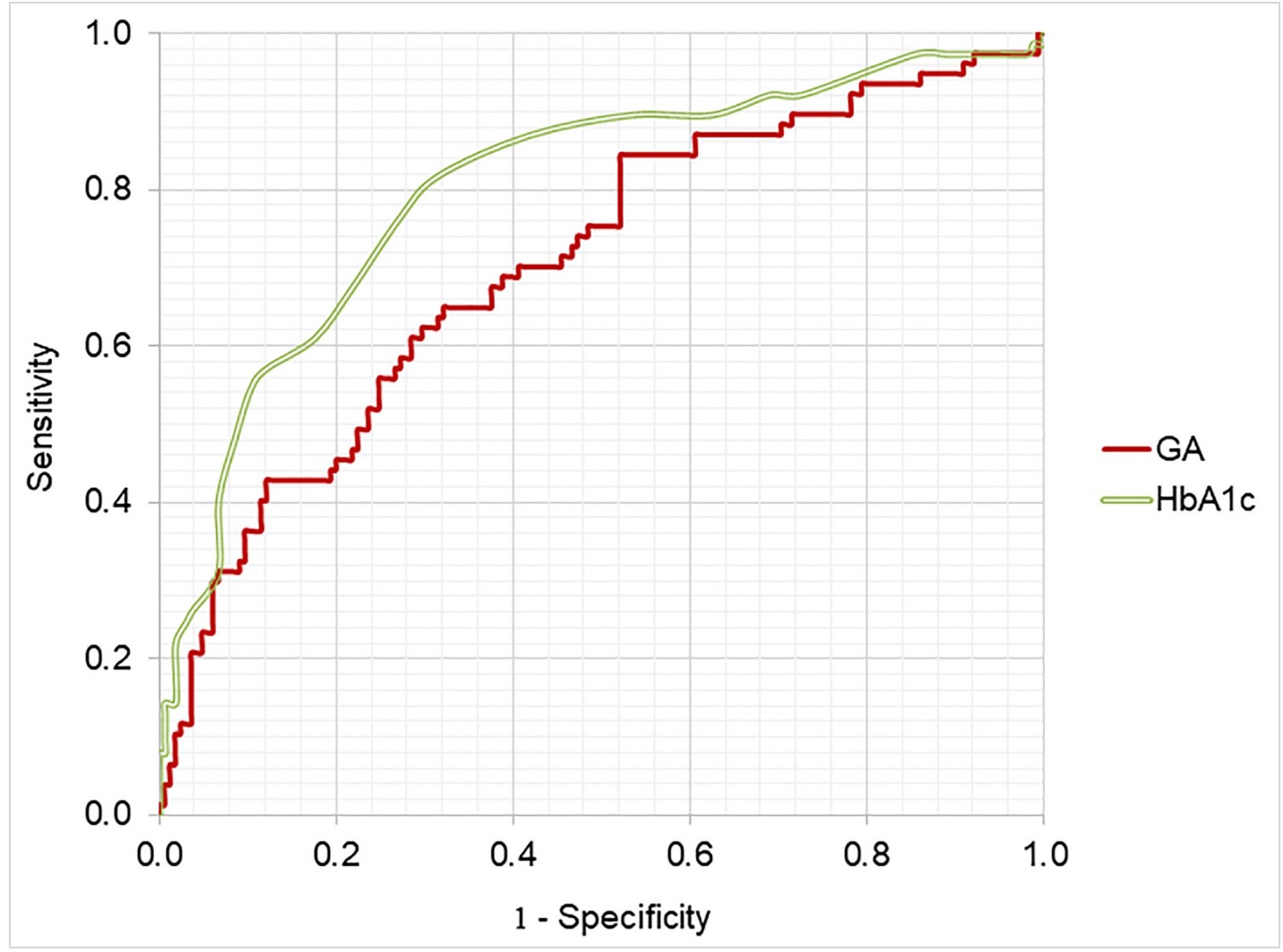

**Fig 2. Receiver operating characteristic (ROC) curves to access the performance of GA, and HbA$_{1c}$ in the diagnosis of DM by OGTT. The AUC value for GA was 0.703 (SE: 0.037, 95% CI: 0.631–0.775, p <0.001) and for HbA$_{1c}$ was 0.802 (SE: 0.032, 95% CI: 0.740–0.864, P <0.001); (n = 242).** HbA$_{1c}$, glycated haemoglobin; GA, glycated albumin; OGTT, oral glucose tolerance test; AUC, area under the ROC curve; SE, standard error; CI, confidence interval.

≥16.8% and another 13/66 were identified only by HbA$_{1c}$ ≥6.5%. A GA ≥16.8% would also identify 14 subjects that would not be detected neither by HbA$_{1c}$ nor OGTT.

## Discussion

This study evaluated the performance of GA test in the diagnosis of DM in Brazilians subjects. According to ROC analysis GA ≥14.8% was the equilibrium cut-off. The LR+ and LR- indicate that GA ≥14.8% is more likely to occur in people with the disease than in people without the disease. However, this cut-off did not show enough sensitivity to correctly define the proportion of people with DM, nor had high enough specificity to correctly define the proportion of people without the disease. On the other hand, GA value of 16.8% presented lower sensitivity but specificity over to 90%, with performance similar to HbA$_{1c}$ ≥6.5% (>48mmol/mol) for detecting DM, and therefore it may be an adequate cut-off point for detecting DM in individuals with high-risk of developing the disease.

**Table 2. Performance of different cut-offs of GA and HbA_{1c} to diagnose DM by OGTT. (n = 242).**

|  | Threshold | Sensitivity (%) | Specificity (%) | LR+ | LR- |
|---|---|---|---|---|---|
| GA (%) | 13.0 | 93.5 | 15.2 | 1.10 | 0.43 |
|  | 14.0 | 84.4 | 44.2 | 1.51 | 0.35 |
|  | 14.8 | 64.9 | 65.5 | 1.88 | 0.54 |
|  | 15.0 | 62.3 | 69.7 | 2.06 | 0.54 |
|  | 15.5 | 48.1 | 77.6 | 2.14 | 0.67 |
|  | 16.0 | 42.9 | 84.8 | 2.83 | 0.67 |
|  | 16.6 | 36.4 | 90.3 | 3.75 | 1.41 |
|  | 16.8 | 31.2 | 93.3 | 4.68 | 0.74 |
|  | 17.0 | 29.9 | 93.9 | 4.93 | 0.74 |
|  | 17.5 | 20.8 | 96.4 | 5.71 | 0.82 |
| HbA_{1c} (%) [mmol/mol] | 5.5 (37.0) | 87.0 | 58.2 | 2.08 | 0.22 |
|  | 5.7 (39.0) | 81.8 | 68.5 | 2.59 | 0.27 |
|  | 5.8 (40.0) | 76.6 | 72.7 | 2.81 | 0.32 |
|  | 6.00 (42.) | 61.0 | 82.4 | 3.47 | 0.47 |
|  | 6.5 (48.0) | 31.2 | 93.3 | 4.68 | 0.74 |
|  | 6.8 (51.0) | 22.1 | 98.2 | 12.14 | 0.79 |

DM, Type 2 diabetes mellitus; HbA_{1c}, glycated haemoglobin; GA, glycated albumin; LR, likelihood ratio; OGTT, oral glucose tolerance test.

Although GA is not currently recommended for the screening or diagnosis of DM, there are several studies which advocate GA as a screening test for undiagnosed DM, still some studies have recommended the test as a secondary screening tool [12–18]. The cut-off of GA ≥16.8% suggested in this study is similar to those proposed by other previous studies [12, 15].

Though GA data have been accumulating in Asian population [12–15], limited data are available in other regions. There are few studies with non-Asians that report on the validity of GA test in screening and diagnosis of DM [16–18]. One study [16] evaluated the performance of GA in obese youth mainly Hispanic North Americans and suggested GA ≥12% as the cut point when using 2hPG as a reference test and GA ≥14% when HbA_{1c} is a reference test. Although no details of sensitivity and specificity were reported, GA was good predictor of DM with AUC >0.90 in both scenarios. Another study evaluated the performance of GA in Caucasian subjects from Italy [17] using HbA_{1c} only as a reference test and reported that the optimal threshold value (GA >14.0%) had sensitivity of 72.2% and specificity of 71.8% for diagnosis of DM. Lately, a study which examined African subjects [18] using OGTT as reference test referred the optimal cut-off value for GA as 14.9%, similar to the optimal threshold of GA of 14.8% reported in this present study. However, it should be noticed that, in this African study the suggested point in the ROC curve is not the point with the best equilibrium between sensitivity and specificity, as sensitivity and specificity for this GA threshold were 64.8% and 93.5%, respectively.

Nevertheless, in comparison with these studies our data showed different sensitivity and specificity for the same cut-off values. Some factors may be related to these differences. Firstly, ethnic differences are an important reason, since GA levels may vary with race/ethnicity independently of glycaemia [26]. Secondly, the inclusion criteria may have an effect in GA performance, our study included Brazilian subjects who had known risk factors for DM presenting in a tertiary hospital, while other studies [12, 13] included subjects from general population.

In this study, only 31.2% (24/86) of diabetic individuals overlapped in the diagnosis of DM by both HbA_{1c} ≥6.5% (≥48mmol/mol) and OGTT criteria. This confirms that there is a large

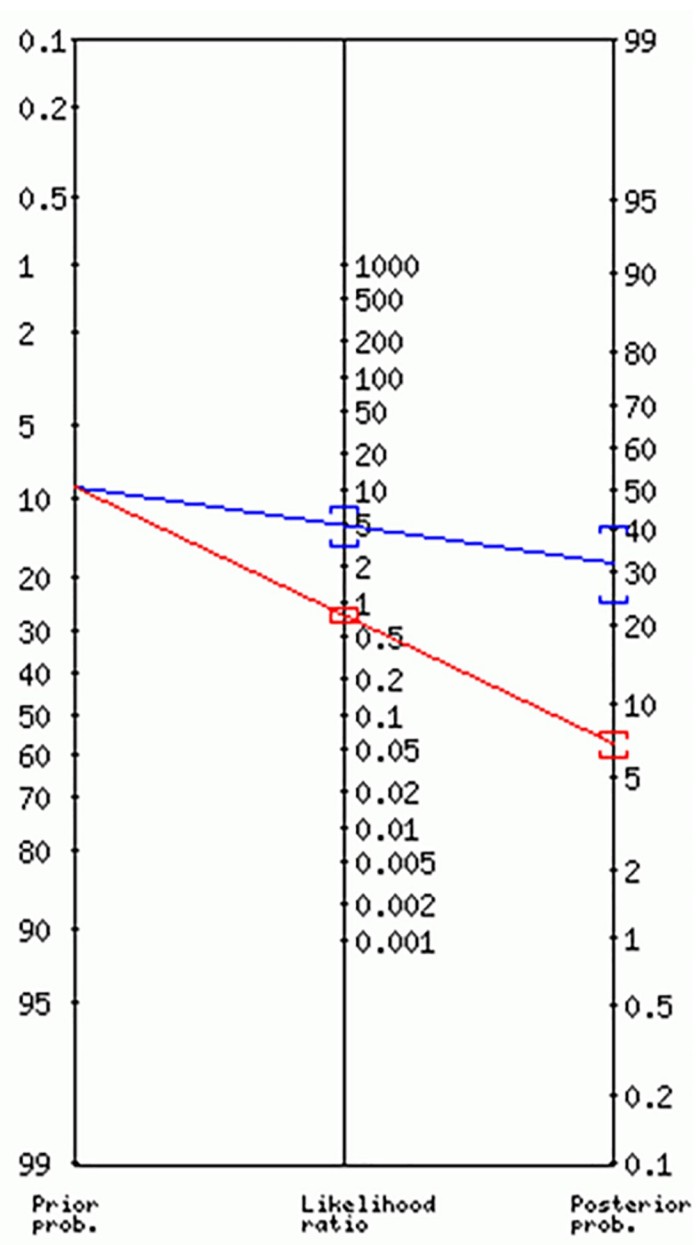

**Fig 3. Fagan´s Nomogram for GA ≥16.8% cut-off inferring a subject's pre- and post-test probability of having DM.** Pre-test probability according to International DM Federation–IDF data [1]; (n = 242).

gap between $HbA_{1c}$ and OGTT criteria for diagnosing diabetes [27]. Our data showed that GA ≥16.8% shows performance similar to $HbA_{1c}$ and detected also one third of diabetic individuals detected by OGTT. Nevertheless, $HbA_{1c}$ and GA do not necessarily detect the same people.

However, using GA would have advantage over $HbA_{1c}$, because GA can be measured accurately in plasma or serum samples [12]. Consequently, GA could be analysed together with common biological markers, including glucose, cholesterol, triglycerides and creatinine, without requiring a blood collection in a separate tube, by contrast, $HbA_{1c}$ can only be measured in whole blood samples. One should be aware that as for $HbA_{1c}$, it is important to recognize that GA is an indirect measure of blood glucose levels and other factors may impact glycation of

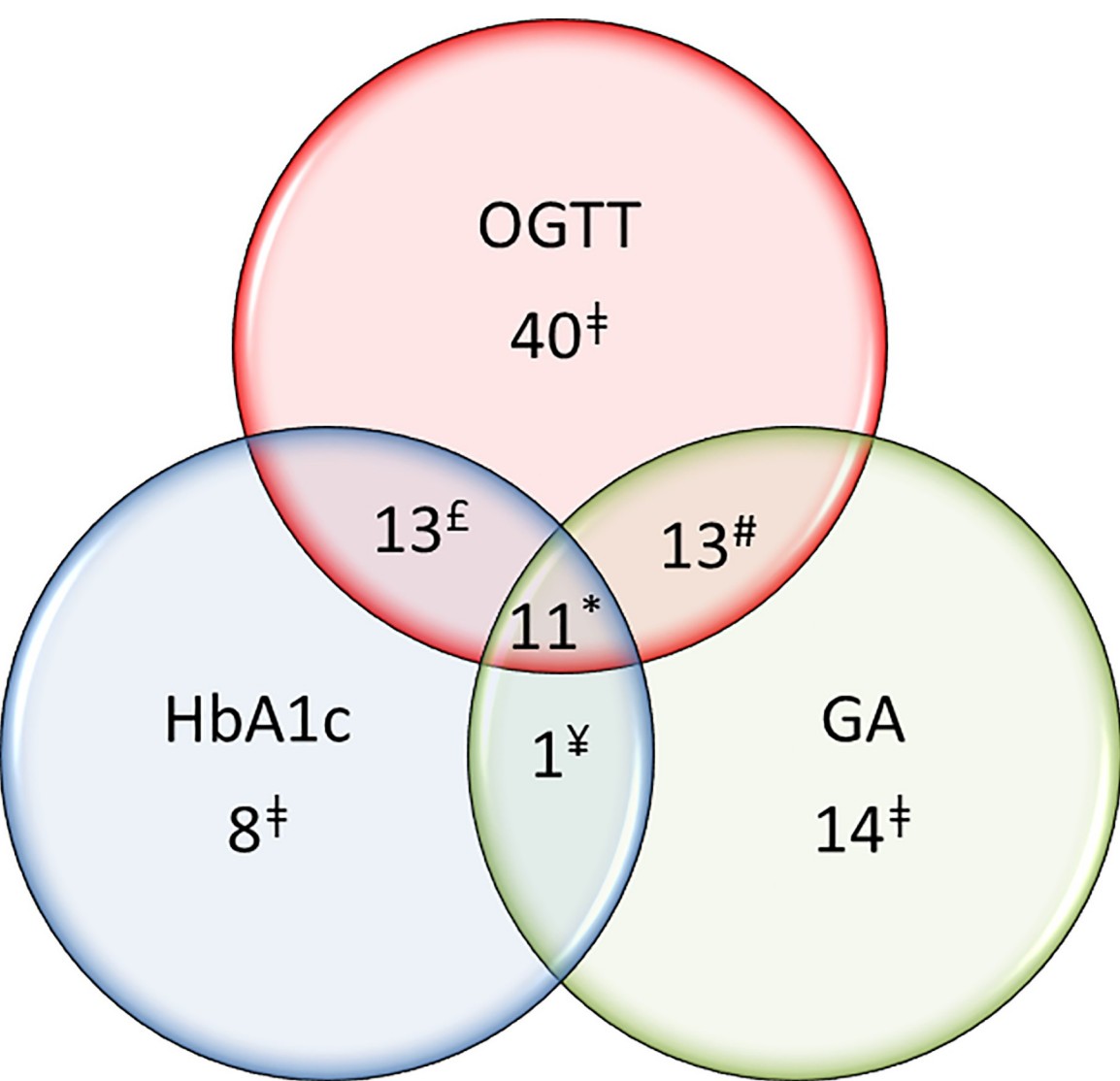

**Fig 4. Number of individuals diagnosed with DM by each test (OGTT, HbA1c, GA) and overlaps. Glycaemic status for HbA1c and OGTT according to ADA criteria [3], and GA >16.8%; (n = 242).** HbA1c, glycated haemoglobin; GA, glycated albumin; OGTT, oral glucose tolerance test; ‡ number of individuals with DM diagnosed by one test criteria without overlapping of other test criteria; * number of individuals with DM diagnosed by all tests (GA ≥16.8%, HbA1c and OGTT) criteria overlapped; £ number of individuals with DM diagnosed by both HbA1c and OGTT criteria overlapped; # number of individuals with DM diagnosed by both GA ≥16.8% and OGTT criteria overlapped; ¥ number of individuals with DM diagnosed by both HbA1c and GA ≥16.8% criteria overlapped.

albumin independently of glycemia status. Therefore, in conditions with altered albumin metabolism as liver cirrhosis, thyroid dysfunction, nephrotic syndrome with massive protein-uria, or inflammatory conditions, the use of GA may be misleading [7, 8]. Other interfering situations on GA levels already described are age and obesity [28, 29].

In the present study, GA and HbA1c were found to be associated with age. A similar association was observed in other studies [12; 13]. However, in a previous analysis from our group, when participants of another study [22] were grouped according to quartiles of age or decade of life there was no difference in GA levels among groups (data not published).

GA was inversely correlated with triglycerides. GA negatively correlated with BMI, WC and LDL, although this association failed to reach statistical significance. Different of GA, HbA1c is

more sensitive to BMI and WC, this may also explain why GA identifies a substantial number of non-obese individuals with prediabetes not detected by HbA1c [30]. Nonetheless, the overall similarity of major DM risk factor associations for elevated HbA1c and GA is reassuring and suggests that, in general, elevations in GA are largely being driven by the same pathophysiological processes that act to raise blood glucose concentrations over time [9–11, 31].

This study has several strengths. It is the first to evaluate the diagnostic utility of GA for DM in Brazilian population. At enrolment, we excluded pregnant women, as well as individuals with anaemia, renal failure, rheumatic disorder, hepatic cirrhosis, or thyroid disease, as these conditions may interfere with the interpretation of HbA1c and GA [7, 8]. Therefore, we were able to evaluate the diagnostic efficacy of GA and HbA1c in the absence of confounding factors. Moreover, the majority of population in this study has European ancestry which allows the applicability of our results in similar populations.

There were also some limitations to the present study that must be considered when interpreting the results. First, the study sample size is small; however, the sample size was calculated a priori and it is sufficient to obtain an AUC of 0.70 with a power of 80% and an estimated alfa error of 5%. Secondly, it comprises mainly individuals at risk of DM with a high pre-test probability attending a tertiary hospital rather than a general population. Third, OGTT, HbA1c and GA were performed only once, even when the results were positive.

## Conclusions

In this study we were able to demonstrate that GA presents overall diagnostic accuracy similar to HbA1c in the diagnosis of DM. Although GA ≥16.8% has comparable performance for diagnosing DM as HbA1c ≥6.5% (>48mmol/mol), GA, HbA1c and OGTT tests do not necessarily detect DM in the same individuals. GA should be used as an additional test rather than an alternative to HbA1c or OGTT and its use as the sole DM diagnostic test should be interpreted with caution to assure the correct classification of diabetic individuals.

## Supporting information

**S1 Table. Correlations of GA, HbA1c and factors potentially associated with the measurement of serum GA in all participants.** [a] Correlation is significant at the 0.01 level (2-tailed). [b] Correlation is significant at the 0.05 level (2-tailed). GA, glycated albumin; HbA1c, glycated haemoglobin; FPG, fasting plasma glucose; 2hPG, plasma glucose 2 h after oral glucose; BMI, body mass index; WC, waist circumference (cm); Trigl., Triglyceride; HDL, serum high density lipoprotein cholesterol; LDL, serum low density lipoprotein cholesterol.
(DOCX)

## Acknowledgments

The authors thank the staff of Clinical Biochemistry Unit and Laboratory Diagnosis Division of Hospital de Clinicas de Porto Alegre for their professionalism and dedication during blood collections and technical assistance.

## Author Contributions

**Conceptualization:** Fernando Chimela Chume, Joíza Lins Camargo.

**Data curation:** Fernando Chimela Chume, Mayana Hernandez Kieling, Gabriela Cavagnolli.

**Formal analysis:** Fernando Chimela Chume, Priscila Aparecida Correa Freitas, Gabriela Cavagnolli, Joíza Lins Camargo.

**Funding acquisition:** Joíza Lins Camargo.

**Investigation:** Fernando Chimela Chume, Mayana Hernandez Kieling, Joíza Lins Camargo.

**Methodology:** Fernando Chimela Chume, Joíza Lins Camargo.

**Project administration:** Fernando Chimela Chume, Mayana Hernandez Kieling, Joíza Lins Camargo.

**Resources:** Joíza Lins Camargo.

**Supervision:** Fernando Chimela Chume, Joíza Lins Camargo.

**Validation:** Fernando Chimela Chume, Priscila Aparecida Correa Freitas, Gabriela Cavagnolli, Joíza Lins Camargo.

**Visualization:** Fernando Chimela Chume, Joíza Lins Camargo.

**Writing – original draft:** Fernando Chimela Chume, Joíza Lins Camargo.

**Writing – review & editing:** Fernando Chimela Chume, Mayana Hernandez Kieling, Priscila Aparecida Correa Freitas, Gabriela Cavagnolli, Joíza Lins Camargo.

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
