## [Decision Letter · Decision Letter 0]

24 Sep 2019

PONE-D-19-21107

Glycated albumin as a diagnostic tool in diabetes: an alternative or an additional test?

PLOS ONE

Dear Dr. Camargo,

Thank you for submitting your manuscript to PLOS ONE. After careful consideration, we feel that it has merit but does not fully meet PLOS ONE’s publication criteria as it currently stands. Therefore, we invite you to submit a revised version of the manuscript that addresses the points raised during the review process.

ACADEMIC EDITOR: Minor revisions requested

We would appreciate receiving your revised manuscript by Nov 08 2019 11:59PM. To enhance the reproducibility of your results, we recommend that if applicable you deposit your laboratory protocols in protocols.io, where a protocol can be assigned its own identifier (DOI) such that it can be cited independently in the future. For instructions see: http://journals.plos.org/plosone/s/submission-guidelines#loc-laboratory-protocols

We look forward to receiving your revised manuscript.

Kind regards,

Petter Bjornstad

Academic Editor

PLOS ONE

Journal Requirements:

Additional Editor Comments (if provided):

Reviewers' comments:

Reviewer's Responses to Questions

**Comments to the Author**

1. Is the manuscript technically sound, and do the data support the conclusions?

Reviewer #1: Yes

Reviewer #2: Yes

2. Has the statistical analysis been performed appropriately and rigorously? 

Reviewer #1: Yes

Reviewer #2: No

3. Have the authors made all data underlying the findings in their manuscript fully available?

Reviewer #1: Yes

Reviewer #2: Yes

4. Is the manuscript presented in an intelligible fashion and written in standard English?

Reviewer #1: Yes

Reviewer #2: Yes

5. Review Comments to the Author

Reviewer #1: The manuscript entitled “Glycated albumin as a diagnostic tool in diabetes: an alternative or an additional test?” by Chume et al. presents an evaluation of the diagnostic use of glycated albumin (GA) as a diagnostic tool for type 2 diabetes in a cohort of individuals from a single tertiary center in Brazil. IRB approval was endorsed and a participant consent process was completed for participation in this study. Overall, the study was well-developed and detailed. The experiments were well-conducted and the analysis was appropriate to evaluate the stated main study question.

TITLE and ABSTRACT:

1. As with the remainder of the paper, would recommend changing all notations of “patients” to either “participants” or “subjects” to follow people first language.

2. It would be helpful to characterize your study population in the results section of the abstract, if possible (i.e. number of participants, average age, BMI, etc.) to allow the reader a frame of reference.

3. Your conclusion statement that GA should be used as an adjunctive test instead of an alternative test to HbA1c or OGTT is somewhat confusing as you state in your introduction that either fasting plasma BG, HbA1c, or 2h OGTT can be used to diagnose T2D – why is GA different than HbA1c if you have come to the conclusion that “GA showed overall diagnostic accuracy similar to HbA1c in the diagnosis of DM” – how did you come to that conclusion?

4. In your abbreviations, I would label DM as "type 2 diabetes mellitus" instead of simply "diabetes mellitus" as you do not refer to other forms of diabetes mellitus such as type 1 diabetes mellitus in this manuscript. You should also define OGTT as an "oral glucose tolerance test" in this section.

INTRODUCTION:

1. Would consider splitting the first paragraph into two paragraphs given the paragraph length.

2. Would recommend adding a hypothesis statement in addition to the aim statement at the end of the “Introduction” section.

METHODS:

1. Why was HbA1c used only for descriptive purposes and comparison with GA if ADA criteria state that T2D can be diagnosed with a FPG ≥126 mg/dL, 2hG on OGTT ≥200 mg/dL, HbA1c ≥6.5%, or random plasma glucose ≥200 mg/dL with symptoms of hyperglycemia? Should it also be used to define participants with T2D if going by true ADA T2D diagnosis criteria? It seems like you use it to diagnose T2D in the results section (i.e. table 2) so the methods section should reflect that.

RESULTS:

1. The average BMI in this paper was noted to 28.9 +/- 6.3 kg/m2 which falls in the overweight category with a large percentage of study participants also being obese, how do you think that impacts your results for GA utility as you have rightly previously stated that both age and obesity are factors that impact GA levels?

2. It’s unclear what separating out clinical and laboratory characteristics of the cohort by the upper tertile of GA values adds to the data given you only reference the equilibrium threshold of 14.8% and the value of 16.8% as the cut off that demonstrates a similar sensitivity/specificity as HbA1c. Why did you select 16.0%? Would it make more sense to select 14.8% or 16.8%?

DISCUSSION:

1. You mention briefly that the HbA1c, GA, and OGTT tests do not reflect the same participants when a diagnosis is made of T2D. I think this is a really important point and it would be good to explore that more as the use of OGTT and HbA1c are currently both accepted for a diagnosis of T2D even though in your population, they only overlapped in terms of a diagnosis of T2D by both measures in 24/86 of the participants. Why do you think that is? It seems like that number would only decrease if combining HbA1c, OGTT, AND GA so what comments do you have about why all three of these measures are detecting T2D in completely different individuals? And does that mean that we should accept a diagnosis of T2D if any one of these tests is positive or if all 3 are positive? Or is one test superior to all of the others? If adding GA as an adjunctive test to the diagnosis of T2D, how would we interpret positive vs. negative results in terms of our diagnosis and management?

CONCLUSIONS:

1. What comments do you have about the generalizability of these results as this study was completed at a single center in Brazil?

2. It would also be worth mentioning that because this study was completed in a population at high risk (i.e. they were referred for an OGTT due to some predisposing factor), results about GA can only really be interpreted if obtained in a similar high-risk population (i.e. one with a high pre-test probability) rather than as a general population screening tool.

Reviewer #2: The authors examined the utility of glycated albumin as a screening tool for diabetes mellitus. Similar work has been performed in other (mostly Asian) populations, and found GA to be useful in some conditions. This report verifies the potential utility of GA as a secondary screening measure in Brazilian patients with high risk of developing DM who were seen at a tertiary hospital. It is promising work in an important area.

My biggest concern is that this is a study cohort with a relatively broad age range, and GA is well known to increase with age. Therefore, it might be worth adjusting these analyses for age, or investigating different cut points for different age groups.

Below are my other comments:

1. In the abstract results (line 43), please include the sensitivity and specificity information for the HbA1c cutoff, so readers can easily compare it to the reported GA cutoff.

2. When comparing paired areas under the ROC curve (e.g. two diagnostic tests on the same group), it is better to use DeLong’s test rather than t-tests.

3. Post-test probability should be calculated directly rather than estimated graphically.

4. Please explain why in table 1 the group is divided by upper tertile of GA, rather than at the equilibrium cut point 14.8% or high specificity cut point 16.8%.

5. Ethnic difference between groups should be assessed with Fisher’s exact test rather than chi-squared, though this is unlikely to make much of a difference. Please also remove the “trended toward significance” language.

6. Line 242 – Please clarify that some of the other studies (e.g. Chan et al.) also suggest GA as a secondary screening tool, not as a primary diagnostic test.

7. Figure 2 – it looks to me as if the HbA1c curve has been smoothed but the GA has not, though this might not be correct.

8. Figure 3 – I find this figure confusing. Please either clarify the legend or remove the figure.

9. Figure 4 – I don’t think this figure is necessary, but it doesn’t detract from the paper either.

10. Please include all CIs in the text as well as figures.

6. PLOS authors have the option to publish the peer review history of their article (what does this mean?). If published, this will include your full peer review and any attached files.

Reviewer #1: No

Reviewer #2: Yes: Tim Vigers

---

## [Author Response · Author response to Decision Letter 0]

22 Oct 2019

ANSWERS TO REVIEWERS:

Reviewer #1: The manuscript entitled “Glycated albumin as a diagnostic tool in diabetes: an alternative or an additional test?” by Chume et al. presents an evaluation of the diagnostic use of glycated albumin (GA) as a diagnostic tool for type 2 diabetes in a cohort of individuals from a single tertiary center in Brazil. IRB approval was endorsed and a participant consent process was completed for participation in this study. Overall, the study was well-developed and detailed. The experiments were well-conducted and the analysis was appropriate to evaluate the stated main study question.

Thank you for your careful evaluation and relevant suggestions/comments.

TITLE and ABSTRACT:

1. As with the remainder of the paper, would recommend changing all notations of “patients” to either “participants” or “subjects” to follow people first language.

Answer: We amended the text accordingly throughout the manuscript. 

2. It would be helpful to characterize your study population in the results section of the abstract, if possible (i.e. number of participants, average age, BMI, etc.) to allow the reader a frame of reference.

Answer: The text was amended in Page 2, lines 39 and 40.

3. Your conclusion statement that GA should be used as an adjunctive test instead of an alternative test to HbA1c or OGTT is somewhat confusing as you state in your introduction that either fasting plasma BG, HbA1c, or 2h OGTT can be used to diagnose T2D – why is GA different than HbA1c if you have come to the conclusion that “GA showed overall diagnostic accuracy similar to HbA1c in the diagnosis of DM” – how did you come to that conclusion?

Answer: Thank you for this comment. The GA cut-off of 16.8% had similar performance for detecting DM as defined by HbA1c >6.5% (>48 mmol/mol) with sensitivity of 31.2% and specificity of 93.3%. As shown in Venn diagram (Figure 3), among 77 subjects diagnosed with DM by OGTT, only 11 were identified with DM by both GA ≥16.8% and HbA1c ≥6.5%. However, 13/77 individuals were identified only by GA ≥16.8% and another 13/77 were identified only by HbA1c ≥6.5%. We amended the text for clarity and change to “GA detected different subjects with DM from those detected by HbA1c, though GA it showed overall diagnostic accuracy similar to HbA1c in the diagnosis of DM” in the Conclusion on page 2, lines 48 and 49. We also elucidate in the results mentioning “Among 77 subjects diagnosed with DM by OGTT, only 11 were identified as DM subjects by both GA ≥16.8% and HbA1c ≥6.5%. Thirteen of 77 subjects were identified only by GA ≥16.8% and another 13/77 were identified only by HbA1c ≥6.5%” on page 17, lines 272 - 275.

4. In your abbreviations, I would label DM as "type 2 diabetes mellitus" instead of simply "diabetes mellitus" as you do not refer to other forms of diabetes mellitus such as type 1 diabetes mellitus in this manuscript. You should also define OGTT as an "oral glucose tolerance test" in this section.

Answer: We amended the text accordingly throughout the manuscript.

INTRODUCTION:

1. Would consider splitting the first paragraph into two paragraphs given the paragraph length.

Answer: We amended the text accordingly. Page 4, line 70.

2. Would recommend adding a hypothesis statement in addition to the aim statement at the end of the “Introduction” section.

Answer: We added the following sentence on page 5, in paragraph 2, lines 99 – 101: “We hypothesized that GA may be used in the diagnosis of DM and in clinical conditions where the HbA1c test does not accurately reflect blood glucose concentrations GA may be an alternative marker”.

METHODS:

1. Why was HbA1c used only for descriptive purposes and comparison with GA if ADA criteria state that T2D can be diagnosed with a FPG ≥126 mg/dL, 2hG on OGTT ≥200 mg/dL, HbA1c ≥6.5%, or random plasma glucose ≥200 mg/dL with symptoms of hyperglycemia? Should it also be used to define participants with T2D if going by true ADA T2D diagnosis criteria? It seems like you use it to diagnose T2D in the results section (i.e. table 2) so the methods section should reflect that.

Answer: Thank you for this comment. This study followed the STARD 2015 reporting guideline for diagnostic accuracy studies which recommend to choose as reference standard test the best available method for establishing the presence or absence of the target condition. OGTT is a direct test for reflection of blood glucose level and numerous studies have shown that it diagnoses most individuals with DM compared with FPG and HbA1c. Besides, HbA1c has low sensitivity at the designated diagnostic cut point. Moreover, the purpose of the present study was to evaluate GA in the diagnosis of DM assessing the ability to be one HbA1c alternative test. However after your comment, we performed an analysis to evaluate the performance of GA using OGTT and/or HbA1c as DM diagnostic reference test. There was no relevant change in the AUC of GA compared to the one obtained when OGTT solely is DM diagnostic reference test [0.708 (95% CI 0.639 – 0.776)] versus [0.703 (95% CI 0.631 – 0.775)], respectively. The optimal cut-off value for serum GA, when OGTT and/or HbA1c are reference was 14.7% (sensitivity 64.0% and specificity 64.1%) versus 14.8% (sensitivity 64.9% and specificity 65.5%) when OGTT alone is reference. 

We have added to the text the results of ROC analysis evaluating the performance of GA using OGTT and/or HbA1c as DM diagnostic reference test (page 14, lines 250 - 256). Also, we amended the text in the Methods section accordingly in page 9, lines 173 - 177.

RESULTS:

1. The average BMI in this paper was noted to 28.9 +/- 6.3 kg/m2 which falls in the overweight category with a large percentage of study participants also being obese, how do you think that impacts your results for GA utility as you have rightly previously stated that both age and obesity are factors that impact GA levels?

Answer: This is a very interesting question. We have analyzed the correlations between GA and factors potentially associated with its measurement but due to manuscript legth we did not presented previously, now we included these data in S1 Table. Also, we amended the text in the Methods (on page 8, lines 171-172), Results (page 14, lines 224-235) and Discussion (page 20, lines 335-344) sections accordingly. GA was associated with age, albumin and triglycerides. However, correlations of GA with BMI, waist circumference (WC) and LDL were not significant. We believe that the size of the study population may be not large enough to evaluate these relationships. Although, different of GA, HbA1c was more sensitive to BMI and WC. This may also explain why in the study of Sumner et al., 2016, GA identified a substantial number of non-obese individuals with prediabetes not detected by HbA1c. The follow reference was added to the manuscript: “Sumner AE, Duong MT, Bingham BA. Glycated Albumin Identifies Prediabetes Not Detected by Hemoglobin A1c: The Africans in America Study. Clin Chem. 2016; 62 (11) 1524-1532. doi: 10.1373/clinchem.2016.261255” (page 28, lines 496 - 498).

2. It’s unclear what separating out clinical and laboratory characteristics of the cohort by the upper tertile of GA values adds to the data given you only reference the equilibrium threshold of 14.8% and the value of 16.8% as the cut off that demonstrates a similar sensitivity/specificity as HbA1c. Why did you select 16.0%? Would it make more sense to select 14.8% or 16.8%?

Answer: Thank you for this observation. We had hypothesized that GA values in the upper tertile had a high probability to diagnose DM, therefore we choose to divide the subjects using 16% of GA value. However, after your comment, we amended the table and described the clinical and laboratory characteristics of the participants divided by subjects with and without DM using ADA OGTT criteria (Table 1 in page 12 and 13). Also, we amended the text in the Methods (page 8, lines 169 and 170) and Results (page 10, lines 207-217) sections accordingly.

DISCUSSION:

1. You mention briefly that the HbA1c, GA, and OGTT tests do not reflect the same participants when a diagnosis is made of T2D. I think this is a really important point and it would be good to explore that more as the use of OGTT and HbA1c are currently both accepted for a diagnosis of T2D even though in your population, they only overlapped in terms of a diagnosis of T2D by both measures in 24/86 of the participants. Why do you think that is? It seems like that number would only decrease if combining HbA1c, OGTT, AND GA so what comments do you have about why all three of these measures are detecting T2D in completely different individuals? And does that mean that we should accept a diagnosis of T2D if any one of these tests is positive or if all 3 are positive? Or is one test superior to all of the others? If adding GA as an adjunctive test to the diagnosis of T2D, how would we interpret positive vs. negative results in terms of our diagnosis and management?

Answer: This a very interesting point, thank you for this remark. FPG, 2h PG during 75g OGTT, and HbA1c are equally appropriate for DM diagnostic testing. Therefore, these tests may be used to screen and diagnose DM. Nevertheless, it should be noted that the tests do not necessarily detect DM in the same individuals. The relationships among FPG, 2h PG and HbA1c are imperfect and they reflect glycemia by different mechanisms resulting in detection of hyperglycaemia in different stages. According to National Health and Nutrition Examination Survey (NHANES) data, HbA1c test, with a diagnostic threshold of 6.5%, diagnoses only 30% of the diabetes cases identified collectively using HbA1c, FPG, or 2h PG (Cowie CC, Rust KF, Byrd-Holt DD, et al. Prevalence of diabetes and high risk for diabetes using A1C criteria in the U.S. population in 1988–2006. Diabetes Care 2010;33:562–568). Compared with FPG and HbA1c tests, 2h PG diagnoses more people with DM [Meijnikman AS, De Block CEM, Dirinck E, et al. Not performing an OGTT results in significant underdiagnosis of (pre)diabetes in a high risk adult Caucasian population. Int J Obes 2017;41:1615–1620]. We previously showed that HbA1c at 6.5% cut-off is not enough to diagnose all cases of DM (reference #27 in this manuscript). In this present study, based on exclusively HbA1c ≥6.5% for DM diagnosis, only 31.2% of diabetic subjects were detected collectively with OGTT. 

Considering the above-mentioned findings and the similarities between GA and HbA1c, GA should be considered as an equally appropriate test for DM diagnosis. Like FPG, 2h PG, and HbA1c, GA would not necessarily detect DM in the same individuals. However, as for HbA1c, it is important to recognize that GA is an indirect measure of blood glucose levels and other factors may impact glycation of albumin independently of glycemia. We highlighted this throughout the discussion.

For a better interpretation of GA values, we believe a better understanding of GA role in DM and prevention of diabetes-specific complications should be extensively studied with long-term follow-up.

CONCLUSIONS:

1. What comments do you have about the generalizability of these results as this study was completed at a single center in Brazil?

Answer: Hospital de Clínicas de Porto Alegre is a large tertiary hospital with multiple specialties located in Porto Alegre, Southern Brazil. Since it is a public hospital with priority for patients of Sistema Único de Saúde (SUS, Brazilian public health care system) has become a reference for the state of Rio Grande do Sul and southern Brazil. Furthermore, in our state the majority of population has European ancestry (https://biblioteca.ibge.gov.br/visualizacao/livros/liv63405.pdf). The proportion of Caucasian descendent people is higher than 80% in this region, similar to the proportion of Caucasian descendent people in this study (80.2%). Therefore, we believe our study population represents a sample of southern Brazil population and may correspond to the majority of European population allowing the applicability of our results in similar populations. These data were added to the text in Page 21, lines 353-355).

2. It would also be worth mentioning that because this study was completed in a population at high risk (i.e. they were referred for an OGTT due to some predisposing factor), results about GA can only really be interpreted if obtained in a similar high-risk population (i.e. one with a high pre-test probability) rather than as a general population screening tool.

Answer: We have added and amended the text accordingly in the study limitations in page 21, lines 359 – 361.

Reviewer #2: The authors examined the utility of glycated albumin as a screening tool for diabetes mellitus. Similar work has been performed in other (mostly Asian) populations, and found GA to be useful in some conditions. This report verifies the potential utility of GA as a secondary screening measure in Brazilian patients with high risk of developing DM who were seen at a tertiary hospital. It is promising work in an important area.

My biggest concern is that this is a study cohort with a relatively broad age range, and GA is well known to increase with age. Therefore, it might be worth adjusting these analyses for age, or investigating different cut points for different age groups.

Answer: We made correlations between GA and factors potentially associated with the measurement of serum GA, and presented in S1 Table. GA was found to be associated with age. GA concentrations increased by 0.44% per decade (GA = 12.503 + 0.044 x age). However similar correlation was found between HbA1c and age. We amended the text accordingly on Page 14, lines 224-235 and Page 20, lines 335-344.

1. In the abstract results (line 43), please include the sensitivity and specificity information for the HbA1c cutoff, so readers can easily compare it to the reported GA cutoff.

Answer: The text was amended on Page 2, line 46.

2. When comparing paired areas under the ROC curve (e.g. two diagnostic tests on the same group), it is better to use DeLong’s test rather than t-tests.

Answer: We apologize, we made a mistake by mentioning that the AUC of GA and HbA1c were compared by T-test, we actually used DeLong’s test in MedCalc. The text was amended accordingly on Page 9, line 178 and lines 190-192.

3. Post-test probability should be calculated directly rather than estimated graphically.

Answer: The Fagan nomogram was only used as a visual aid to help interpretation and applicability. All post-test probabilities were calculated by using the following formula: Post-test probability = Post-test odds /(Post-test odds + 1). In this equation, positive post-test probability was calculated using the likelihood ratio positive, and the negative post-test probability was calculated using the likelihood ratio negative. Likelihood ratio was calculated from sensitivity and specificity of GA test. Post-test odds = Pre-test odds x Likelihood ratio, being Pre-test odds = (Pre-test probability /(1 - Pre-test probability). Pre-test probability was according to International Diabetes Federation – IDF data. However, for illustration of the results and better understanding of clinical applicability, we used the Fagan nomogram (Figure 4; where we mentioned the above formulas).

4. Please explain why in table 1 the group is divided by upper tertile of GA, rather than at the equilibrium cut point 14.8% or high specificity cut point 16.8%.

Answer: As mentioned before in our response for Reviewer 1, we had hypothesized that GA values in upper tertile had a high probability to diagnose DM, therefore we choose to divide the subjects using 16% of GA value. However, after reviewers comments, we amended the table and described the clinical and laboratory characteristics of the participants divided by subjects with and without DM using ADA OGTT criteria (Table 1 in page 12 and 13). Also, we amended the text in the Methods (page 8, lines 169 and 170) and Results (page 10, lines 207-217) sections accordingly.

5. Ethnic difference between groups should be assessed with Fisher’s exact test rather than chi-squared, though this is unlikely to make much of a difference. Please also remove the “trended toward significance” language.

Answer: We re-analyzed these different by Fisher´s exact test and as you said, the results were similar (p = 0.147 vs. 0.152 for Fisher’s exact and chi-squared test, respectively). However, we amended accordingly (Table 1 in page 12 and 13). Also, we amended the text in the Methods section (page 8, line 168), and removed the “tended to be significant” on page 10, lines 210-211.

6. Line 242 – Please clarify that some of the other studies (e.g. Chan et al.) also suggest GA as a secondary screening tool, not as a primary diagnostic test.

Answer: We amended the text for clarity and change to “Although GA is not currently recommended for the screening or diagnosis of DM, there are several studies which advocate GA as a screening test for undiagnosed DM, still some studies have recommended the test as a secondary screening tool” on page 18, line 290.

7. Figure 2 – it looks to me as if the HbA1c curve has been smoothed but the GA has not, though this might not be correct.

Answer: That is interesting observation, but it is just impression. We believe since HbA1c has few data points (markers) the curve chart gives impression as if has been smoothed, while GA has numerous data points give unsmoothed shape.

8. Figure 3 – I find this figure confusing. Please either clarify the legend or remove the figure.

Answer: We used Venn diagram to present the number of individuals with DM diagnosed by each test and overlaps. We would like to keep the figure to illustrate the relationships between the tests, highlighting how the they are similar and different. We amended the legend for clarity accordingly (page 29, lines 510 – 518).

9. Figure 4 – I don’t think this figure is necessary, but it doesn’t detract from the paper either.

Answer: As we mentioned above, the Fagan nomogram was used as a visual aid to help interpretation and applicability of our findings. Therefore, we believe it is of certain utility for readers and, if possible, we would like to keep this figure for clarity.

10. Please include all CIs in the text as well as figures.

Answer: We revised and amended the manuscript accordingly. 

Last, we would like to express our appreciation to you and the reviewers advices for suggesting how to improve our paper.

Thank you very much!

Yours sincerely

---

## [Decision Letter · Decision Letter 1]

12 Dec 2019

Glycated albumin as a diagnostic tool in diabetes: an alternative or an additional test?

PONE-D-19-21107R1

Dear Dr. Camargo,

We are pleased to inform you that your manuscript has been judged scientifically suitable for publication and will be formally accepted for publication once it complies with all outstanding technical requirements. Congratulations!

With kind regards,

Petter Bjornstad

Academic Editor

PLOS ONE

Additional Editor Comments (optional):

Reviewers' comments:

Reviewer's Responses to Questions

**Comments to the Author**

1. If the authors have adequately addressed your comments raised in a previous round of review and you feel that this manuscript is now acceptable for publication, you may indicate that here to bypass the “Comments to the Author” section, enter your conflict of interest statement in the “Confidential to Editor” section, and submit your "Accept" recommendation.

Reviewer #1: All comments have been addressed

Reviewer #2: All comments have been addressed

2. Is the manuscript technically sound, and do the data support the conclusions?

Reviewer #1: Yes

Reviewer #2: Yes

3. Has the statistical analysis been performed appropriately and rigorously? 

Reviewer #1: Yes

Reviewer #2: Yes

4. Have the authors made all data underlying the findings in their manuscript fully available?

Reviewer #1: Yes

Reviewer #2: Yes

5. Is the manuscript presented in an intelligible fashion and written in standard English?

Reviewer #1: Yes

Reviewer #2: Yes

6. Review Comments to the Author

Reviewer #1: Fantastic work with your responses to questions from the reviewers. A few errors that were noted that should be corrected before the manuscript makes it into print are as follows:

INTRODUCTION:

1. Type 2 DM mellitus (DM) on page 4 line 61 should be changed to type 2 diabetes mellitus (DM) for clarity. This should also be reflected on page 4 line 67 where it should read DM instead of DM type 2 (due to previous definition).

TABLE 1:

1. All fields with mean +/- SD should include a +/- and not just a + in this table.

DISCUSSION:

1. Page 21 line 352 should state "enrollment" instead of "enrolment."

Reviewer #2: (No Response)

7. PLOS authors have the option to publish the peer review history of their article (what does this mean?). If published, this will include your full peer review and any attached files.

Reviewer #1: No

Reviewer #2: No

---

## [Editor Report · Acceptance letter]

18 Dec 2019

PONE-D-19-21107R1 

Glycated albumin as a diagnostic tool in diabetes: an alternative or an additional test? 

Dear Dr. Camargo:

I am pleased to inform you that your manuscript has been deemed suitable for publication in PLOS ONE. Congratulations! Your manuscript is now with our production department. 

With kind regards,

on behalf of

Dr Petter Bjornstad 

Academic Editor

PLOS ONE